# Analysis of Patients with Alcohol Dependence Treated in Silesian Intensive Care Units

**DOI:** 10.3390/ijerph19105914

**Published:** 2022-05-12

**Authors:** Małgorzata Łowicka-Smolarek, Izabela Kokoszka-Bargieł, Małgorzata Knapik, Konstanty Śmietanka, Piotr Dyrda, Mateusz Możdżeń, Magdalena Kurczab, Jarosław Borkowski, Piotr Knapik

**Affiliations:** 1Department of Anaesthesiology and Intensive Therapy, Silesian Centre for Heart Diseases in Zabrze, Medical University of Silesia, 40-055 Katowice, Poland; gosialowicka@o2.pl (M.Ł.-S.); knapikmalgosia1@gmail.com (M.K.); illevi@me.com (K.Ś.); lek.piotrdyrda@gmail.com (P.D.); magdalena.kurczab@gmail.com (M.K.); mjbor1@wp.pl (J.B.); 2Department of Anesthesiology and Intensive Therapy, Provincial Specialist Hospital, 43-100 Tychy, Poland; sek.kan@sccs.pl; 3Students’ Scientific Society, Department of Cardiac Anesthesia and Intensive Care, Medical University of Silesia, 40-055 Katowice, Poland; mateusz.mozdzen7@gmail.com

**Keywords:** alcohol dependency, ICU, mortality, neurological injury

## Abstract

Analysis of patients with alcohol dependence (AD) treated in intensive care units has never been performed in Poland. Data from 25,416 adult patients identified in a Silesian Registry of Intensive Care Units were analysed. Patients with AD were identified, and their data were compared with the remaining population. Preadmission and admission variables that independently influenced ICU death in these patients were identified. Among 25,416 analysed patients, 2285 subjects (9.0%) were indicated to have AD among their comorbidities. Patients with AD were significantly younger (mean age: 53.3 ± 11.9 vs. 62.2 ± 15.5 years, *p* < 0.001) but had a higher mean APACHE II score at admission and were more frequently admitted to the ICU due to trauma, poisonings, acute pancreatitis, and severe metabolic abnormalities. ICU death and unfavourable outcomes were more frequent in these patients (47.8% vs. 43.0%, *p* < 0.001 and 54.1% vs. 47.0%, *p* < 0.001, respectively). Multiorgan failure as the primary cause of ICU admission was among the most prominent independent risk factors for ICU death in these patients (OR: 3.30, *p* < 0.001). Despite the younger age, ICU treatment of patients with AD was associated with higher mortality and a higher percentage of unfavourable outcomes.

## 1. Introduction

Disorders associated with alcohol use pose a tremendous burden to healthcare systems around the world [1]. Overall, 5.1% of the global burden of disease and injury is attributable to alcohol, as measured in disability-adjusted life years (DALYs) [2]. According to the data from the literature, around 25% of patients admitted to intensive care units (ICUs) might have alcohol-related issues, and AD may be responsible for numerous clinical problems encountered in the ICU setting [3].

It has been known for a long time that patients with AD create a special population in ICU. AD is known to be independently associated with the frequency of sepsis and septic shock among adult ICU patients [4]. Chronic alcohol abuse significantly increases the risk of developing ARDS [5]. Alcoholic patients have increased morbidity and mortality in postoperative periods [6]. They also experience longer ICU and hospital length of stay, greater resource utilisation, and significantly increased mortality, compared with the remaining ICU population [7]. For all these reasons, the population of patients with AD deserves special attention in the ICU setting.

Moreover, patients with AD are frequently admitted to ICUs. The percentage of these patients varies across countries and ranges from 7% to 33% [8]. Despite this fact, data on ICU treatment of these patients are relatively scarce [1,9,10,11]. At present, there are no data on ICU treatment of AD patients in Poland. To the best of our knowledge, no studies based on a comparable number of data, analysing the impact of alcohol dependence on survival in ICU patients, have been published so far in the medical literature, and the majority of the already existing reports are single-centre studies.

The primary aim of our study was to compare admission variables, the course of treatment, and the outcomes of ICU patients with AD with the remaining ICU population. The secondary aim was to identify independent variables affecting ICU death in this specific population.

## 2. Materials and Methods

This retrospective, multicentre study is based on the data from the Silesian Registry of Intensive Care Units. The dataset included anonymous, medical variables of 25,416 ICU hospitalisations. These hospitalisations occurred in the ICUs located in the Silesian Region of Poland between 1 October 2010 and 31 December 2019. Approximately 35% of all Silesian ICUs reported their data to the registry at that time (the Registry is voluntary). The Silesian region is a post-industrial area covering only 3.9% of Polish territory, but it is inhabited by 11.9% of the Polish population.

In the Registry, there are data on circumstances surrounding ICU admission, patients’ comorbidities, admission diagnoses, as well as the course and results of ICU treatment [12]. Due to the retrospective and anonymous nature of the study, the Ethical Committee at the Medical University of Silesia in Katowice waived the requirement for obtaining consent from patients to participate in the study.

In a group of comorbidities, one of the introduced variables is information about the presence of AD. According to the Registry definitions (available via the Registry website), this variable may be introduced to the dataset if there is clear information on the presence of AD in previous discharge cards or/and on the basis of the information obtained directly from the patient or his family. Once a patient is discharged from the ICU, this information is entered into the system (together with the other data) into a group of variables called “additional comorbidities present before ICU admission”.

All patients with AD were identified and compared with the remaining population treated in the ICU at the same time. Preadmission and admission variables (including demographic parameters, circumstances surrounding ICU admission, comorbidities, and primary reasons for ICU admission), details regarding treatment, and the outcomes were compared. Unfavourable outcome was defined as ICU death or discharge from the ICU in a vegetative state or minimally conscious state.

Additionally, in a subgroup of patients with AD, we aimed to identify independent predictors for ICU death among preadmission and admission variables. This was performed by comparing these variables between non-survivors and survivors of the ICU stay with the use of a univariable analysis. Further, independent variables influencing death in a population of patients with AD were identified with the use of multivariable logistic regression.

Analyses and graphs were performed with the use of Dell Inc. (2016) (nalyses and graphs were performed with the use of TIBCO Software Inc. (2017), Statistica (data analysis software system), version 13. (http://statistica.io, (accessed on 16 February 2022))) Dell Statistica (data analysis software system), version 13. Demographic data were presented using descriptive statistics methods and compared using Student’s *t*-test or Mann–Whitney test. For the comparison of qualitative variables, a chi-squared test with Yates correction was used. The effect of independent variables on the outcome variable of interest was calculated by means of univariable logistic regression. Variables with *p* value < 0.05 were included in multivariable logistic regression analysis. The multivariable model was fitted using the stepwise method, where *p* < 0.05 was set as inclusion and removal criteria. For purposes of all calculations, statistical significance was considered for *p* <0.05.

## 3. Results

In a group of 25,416 hospitalisations, 2285 patients with AD were identified (9.0%). The percentage of male patients was significantly higher among patients with AD (80.6% vs. 54%, *p* < 0.001). Patients with AD were also significantly younger in comparison to the remaining population (mean age: 53.3 ± 11.9 vs. 62.2 ± 15.5 years, *p* < 0.001). The overall number of patients and percentage of patients with AD in various age groups are presented in Figure 1.

Marked differences between patients with AD and the remaining ICU population could be observed in terms of circumstances surrounding ICU admission. AD patients were more often admitted to the ICU directly from the emergency department. Direct admissions from the operating theatre were relatively rare in this group and almost entirely concerned trauma patients (Figure 2).

The analysis of data presented in Table 1 indicates that the population of patients with AD significantly differed from the remaining patients in terms of the comorbidities (Table 1). There were significantly fewer patients with chronic diseases in this group (such as coronary heart disease, arterial hypertension, disseminated atherosclerosis, chronic heart failure, diabetes, etc.). There was also a significantly higher percentage of patients with cachexia on admission (defined as BMI < 15 kg/m^2^).

The distribution of primary reasons for ICU admission was also different in patients with AD (Table 2). A higher percentage of AD patients were admitted to the ICU due to trauma, poisoning, acute pancreatitis, and severe metabolic abnormalities. Moreover, cardiac arrest and the need for cardiopulmonary resuscitation before ICU admission was also more frequent in these patients.

Overall, 54.8% of patients with AD and 51.0% of the remaining patients were assessed with the use of the APACHE II score on admission. APACHE II score was significantly higher among patients with AD (23.9 ± 8.5 vs. 22.9 ± 9.0 points, *p* < 0.001).

Despite all these differences, ICU treatment was largely similar in both groups, with the exception of non-invasive ventilation being rarely used in this group. However, fewer patients with AD underwent sophisticated, non-standard procedures such as plasmapheresis, intra-aortic balloon pump, or extracorporeal membrane oxygenation. Due to the more frequent cardiac arrest before ICU admission, significantly more patients with AD underwent therapeutic hypothermia (Table 3).

ICU mortality among patients with AD was significantly higher in comparison to the remaining population (47.8% vs. 43.5%, *p* < 0.001). When referred to the admission APACHE II scoring system, the expected-to-observed mortality ratio (O/E ratio) was 0.99 for patients with AD and 0.93 for the remaining population. Graphical presentation of the expected and observed mortalities across various ranges of APACHE II scoring results for AD and non-AD patients is shown in Figure 3.

Apart from the higher ICU mortality, more AD patients were also discharged from the ICU with profound neurological damage (6.3% vs. 4.0, *p* < 0.001). Therefore, patients with AD more frequently completed their ICU stay with ICU death (47.8% vs. 43.0%, *p* < 0.001), as well as with composite unfavourable outcomes (54.1% vs. 47.0%, *p* < 0.001).

A comparison of preadmission and admission data of survivors and non-survivors in a group of patients with AD is presented in Table 4 and Table 5. Multivariable analysis carried out on the basis of data listed in these tables revealed that there were 10 independent variables affecting ICU death among patients with AD (Figure 4). Multiorgan failure as the primary cause of ICU admission was among the most prominent independent risk factors for ICU death in these patients (OR: 3.30, 95% CI: 2.50–4.36, *p* < 0.001).

## 4. Discussion

In our study, we found that the outcomes of treatment of alcoholic patients in the ICUs located in the Silesian Region of Poland were significantly worse when compared with the remaining population. This applies to ICU mortality but also to composite unfavourable ICU outcomes (defined as death or discharge with a profound neurological injury). Such results have been observed despite the fact that patients with AD were significantly younger in comparison to the remaining ICU population.

It may be easily observed that ICU mortality confirmed in the Silesian ICU Registry was very high (also among non-AD patients). ICU mortality higher than 40% might be surprising, but it is necessary to remember that mortality rates in Polish ICUs are already known to be higher than in other European countries [13,14]. Based on data available from the Polish National Institute of Public Health, ICU mortality in Poland (for the year 2012) exceeded 42% [15].

In our ICU population, the percentage of patients with AD was 9%. According to the data from the World Health Organisation, AD affects approximately 3.4% of adult Europeans and 5% of the adult population of Poland [16]. It was also found that the percentage of AD patients among ICU admissions in other countries may be much higher. In Finland, as much as 17.5% of hospitalisations at ICU concerned patients with AD [11]. In Scotland, this percentage was even higher and was over 25% [17], while in Australia, it ranged from 21% to 25% [1,18].

Patients with AD in our study were found to be significantly younger than the rest of the population, and the percentage of men was also higher in this group. These results are consistent with the trends observed in the global population. Geary et al. reported that the median age for AD patients admitted to Scottish ICU was only 51 years (in comparison to 63 years for the remaining population) [17]. Based on population studies conducted in Europe, we are also aware of the fact that in the group of almost 11 million people with AD there were approximately three times more men in comparison to women [16]. Alcohol is also known to be a major risk factor for all types of injuries [19]. In Poland, about 40% of deaths in people up to 44 years of age are due to external causes (other than natural). They include poisoning, injuries, traffic accidents, drowning, and suicides, and most of the victims in this age group are men [20]. Considering that alcohol is the main risk factor for injuries and that injuries most often affect young men, it is not surprising that younger male patients predominated in our ICU population with AD.

In our ICU population, patients with AD were most commonly admitted to the ICU from the emergency department. This finding is relatively easy to explain as most of these admissions were emergent and patients were usually not admitted from other hospital departments. Alcohol consumption is known to be a causal factor in a variety of diseases and injuries resulting from violence, road accidents, and collisions [2]. In patients with medical diagnoses associated with ICU admission, a diagnosis of AD is known to be common and is associated with a prolonged duration of mechanical ventilation [21]. These patients are also more likely to be admitted under a diagnostic code of trauma, pancreatitis, and upper gastrointestinal bleeding [1].

In our analysed population with AD, there were significantly fewer patients with chronic diseases (such as coronary heart disease, arterial hypertension, disseminated atherosclerosis, chronic heart failure, diabetes, etc.). This could be again explained by the fact that patients with AD hospitalised in the ICU were significantly younger than the remaining (non-AD) population. This age difference translates into a reduced incidence of chronic diseases, commonly associated with middle-aged and elderly people [22].

It has been noted that there was a strikingly higher percentage of patients with very low BMI values (<15 kg/m^2^) in our ICU population with AD (11.1% vs. 2.8%, *p* < 0.001). This fact may be due to the pathophysiology of alcoholism. Malnutrition is a common problem in these patients. The aetiology of this disorder is dual: primary (resulting from insufficient supply of nutrients as a result of replacing them with alcohol) and secondary (because the nutrients remain metabolically inaccessible due to the digestive and absorption disorders, disturbed utilisation processes, and hepatic degradation). This problem is known to be particularly acute among patients requiring hospitalisation in the course of complications of chronic alcohol abuse [23,24].

The primary causes of hospitalisation in our patients with AD were significantly different from those observed in the remaining population. Patients were mainly admitted to the ICU due to injuries, poisoning, acute pancreatitis, severe metabolic disorders, and cardiac arrest, which is consistent with the data from the literature. We have already revealed that alcohol abuse and AD are significantly related to almost all types and mechanisms of trauma [19], in particular traumatic brain injury [25,26]. Moreover, alcohol intoxication may overestimate the assessment of traumatic brain injury severity at the time of admission [27]. Alcohol still remains the dominant etiological factor of acute pancreatitis, particularly in eastern European countries [28,29]. Severe pancreatitis often requires ICU treatment, which explains the high incidence of this disease in our study population when compared with the control group [30]. Alcohol consumption is also often associated with accidental or intentional poisoning, additionally influencing the morbidity of the study group [31].

A slightly different explanation concerns the higher frequency of severe metabolic abnormalities and cardiac arrest in the subpopulation studied. Patients with AD frequently require hospital admissions (and ICU admissions) due to alcoholic ketoacidosis. In laboratory tests, these patients may present with metabolic acidosis, with an increased anion gap and severe electrolyte disturbances [32]. Alcohol consumption also has a complex effect on the function of the cardiovascular system, and therefore, life-threatening ventricular arrhythmias and sudden cardiac death are relatively frequent in these patients. The aetiology of arrhythmias is complex and includes both the direct influence of alcohol, the prolongation of the QT interval caused directly by ethanol, as well as its interactions with medications and psychoactive substances that may also prolong the QT interval. Hypomagnesaemia and hypokalaemia, common in AD patients, may additionally predispose patients to life-threatening ventricular arrhythmias including torsade de pointes [33].

According to our data, significantly fewer patients with AD underwent sophisticated, non-standard ICU procedures such as plasmapheresis, insertion of an intra-aortic balloon pump, or the application of extracorporeal membrane oxygenation. This is quite understandable. Reasons for admitting AD patients are usually different from the remaining population and usually do not require such invasive procedures. In this group, civilisation diseases are rare, and if they do additionally occur, they significantly reduce patients’ overall chances of survival. In the case of alcoholics, the presence of severe chronic hepatic insufficiency or advanced mental disorders is often sufficient to abandon any invasive procedures [34]. As a result of these circumstances, AD patients are unlikely to be qualified for more sophisticated and invasive ICU procedures.

As presented in the Results section of our study, ICU mortality among patients with AD was significantly higher (despite the fact that this population was younger). This finding is fully supported by other studies. In an extensive review of the medical literature on this topic, Mehta indicated that patients with AD are predisposed to developing withdrawal syndromes and other conditions that may require ICU admission, and they are also characterised by a higher rate of complications, longer ICU and hospital length of stay, greater resource utilisation, and, ultimately, significantly increased mortality [3]. These findings have also been confirmed in many other studies [4,5,6,8,10,17,19,35]. It has also been revealed that in ICU patients admitted with infection, even a moderate alcohol consumption might negatively affect the prognosis [35].

Multivariable analysis revealed that there were several independent variables affecting ICU death in our subgroup of patients with AD. Multiorgan failure as the primary cause of ICU admission was the most prominent independent risk factor for ICU death in these patients. The other independent risk factors included also cachexia (among comorbidities) and cardiac arrest, shock, severe metabolic disorders, intoxication, exacerbation of circulatory failure, and craniocerebral and multiple trauma (among primary causes of ICU admissions) (Figure 4). Unfortunately, it would be very difficult to refer all these results to comparative results in the medical literature. This is due to the fact that these data are derived from the population, which has never been studied before in Poland.

Our study has some significant limitations. This is a retrospective study which is always prone to bias. The Registry is local (limited to the Silesian region of Poland), and only 35% of Silesian ICUs report to this Registry. We used a simplified definition of AD, not based on classical diagnostic criteria; however, only this definition was valid, accepted, and known to the Registry users, when these data have been collected. Therefore, it is possible that in our control group (no AD), we might have included a few patients with alcohol use disorders (DSM 5) and classified as “alcohol abuse” (DMS-IV) or “mild alcohol use disorder” (DSM-V). All these deficiencies, however, are balanced by the large sample size and the significance of the collected data.

## 5. Conclusions

In summary, we conclude that ICU outcomes of alcoholic patients in Poland are poor and significantly worse than in the remaining ICU population. In more than half of these patients, unfavourable outcomes of ICU treatment (defined as death or discharge with profound neurological damage) may be expected. Cardiac arrest before ICU admission is the most prominent independent risk factor for ICU death in this population.

## Figures and Tables

**Figure 1 ijerph-19-05914-f001:**
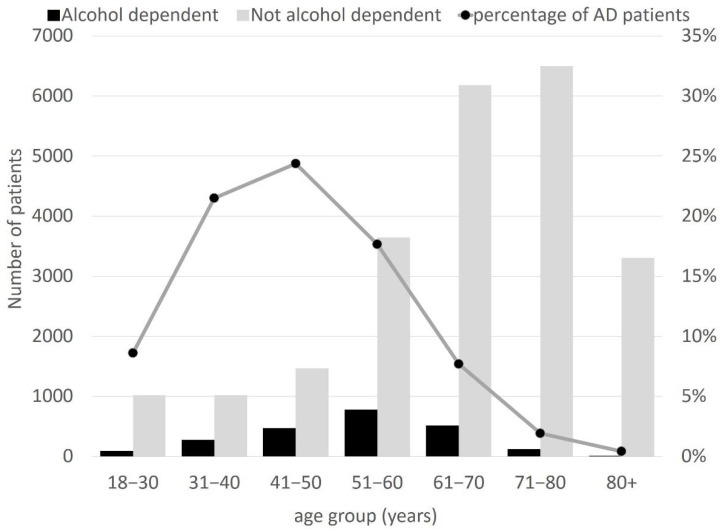
Overall number of patients and the percentage of patients in various age groups.

**Figure 2 ijerph-19-05914-f002:**
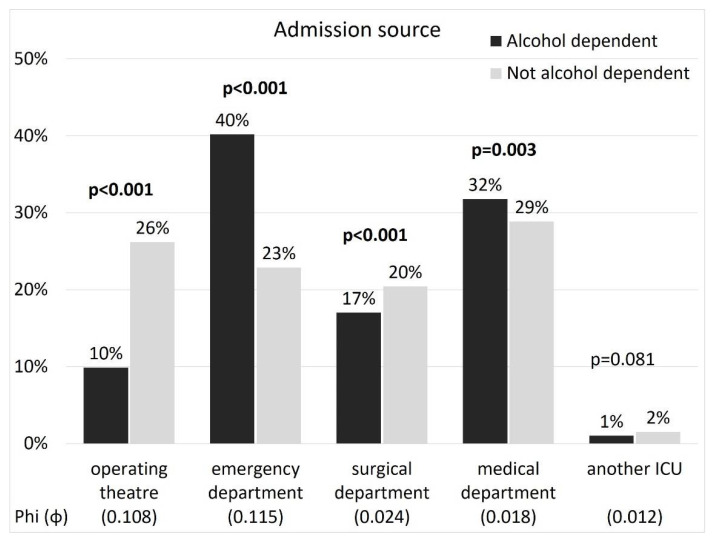
Source of ICU admission in patients with and without AD.

**Figure 3 ijerph-19-05914-f003:**
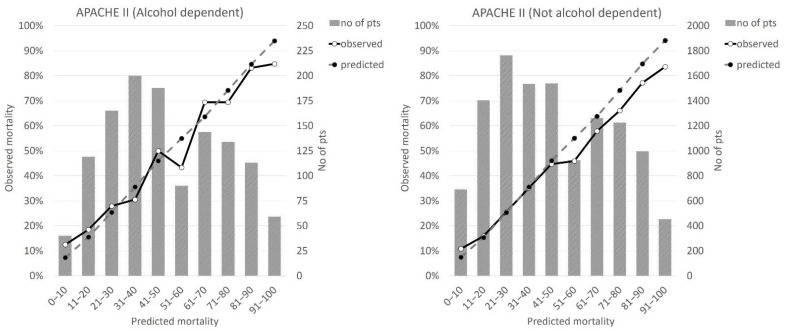
Expected and observed mortalities across various ranges of APACHE II scoring results in patients with and without AD (patients with AD—(**left figure**), patients without AD—(**right figure**)).

**Figure 4 ijerph-19-05914-f004:**
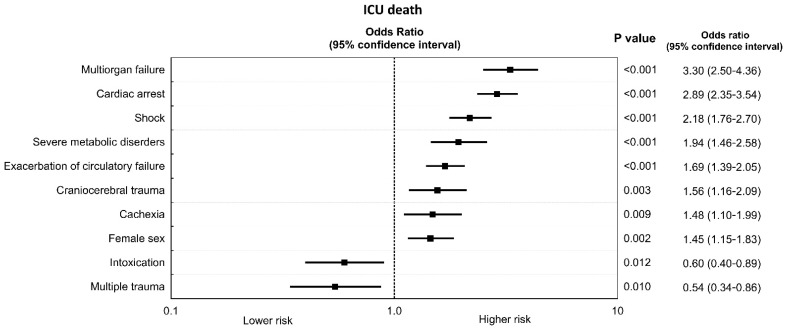
Independent predictors of ICU death in patients with AD.

**Table 1 ijerph-19-05914-t001:** Medical status at ICU admission.

		Alcohol Dependent		Not Alcohol Dependent		*p*
		(n = 2285)		(n = 23,131)		
Admission	First	2189	(95.8%)	21,781	(94.2%)	**0.002**
	Second	92	(4.0%)	1176	(5.1%)	**0.030**
	Another	4	(0.2%)	174	(0.8%)	**0.002**
Co-morbidities	Coronary artery disease	355	(15.5%)	10,137	(43.8%)	**<0.001**
	Heart failure	344	(15.1%)	8500	(36.7%)	**<0.001**
	Arterial hypertension	626	(27.4%)	12,622	(54.6%)	**<0.001**
	Disseminated atherosclerosis	405	(17.7%)	8116	(35.1%)	**<0.001**
	Chronic respiratory failure	147	(6.4%)	3013	(13.0%)	**<0.001**
	Home oxygen therapy	3	(0.1%)	378	(1.6%)	**<0.001**
	Extreme obesity	39	(1.7%)	1417	(6.1%)	**<0.001**
	Cachexia	253	(11.1%)	648	(2.8%)	**<0.001**
	Diabetes	270	(11.8%)	6120	(26.5%)	**<0.001**
	Chronic renal failure	120	(5.3%)	3629	(15.7%)	**<0.001**
	Dialysis dependency	11	(0.5%)	310	(1.3%)	**0.001**
	Previous cerebral stroke	78	(3.4%)	1824	(7.9%)	**<0.001**
	Chronic neurological disorders	219	(9.6%)	1736	(7.5%)	**<0.001**
	Systemic autoimmune diseases	4	(0.2%)	304	(1.3%)	**<0.001**
	Post-transplant	1	(0.0%)	73	(0.3%)	**0.036**
	Cancer	27	(1.2%)	2095	(9.1%)	**<0.001**
	Pregnancy	0	(0.0%)	41	(0.2%)	0.082
	None	0	(0.0%)	2413	(10.4%)	**<0.001**

**Table 2 ijerph-19-05914-t002:** Primary reason for ICU admission among patients with AD and the remaining population.

Variables	Alcohol Dependent		Not Alcohol Dependent		*p*
	(n = 2285)		(n = 23,131)		
Acute respiratory failure	1722	(75.4%)	17,259	(74.6%)	0.448
Exacerbation of resp. failure	82	(3.6%)	1975	(8.5%)	**<0.001**
Exacerbation of circulatory failure	857	(37.5%)	11,053	(47.8%)	**<0.001**
Multiorgan failure	373	(16.3%)	2929	(12.7%)	**<0.001**
Shock	676	(29.6%)	6961	(30.1%)	0.629
Cardiac arrest	717	(31.4%)	5401	(23.3%)	**<0.001**
Disorders of consciousness	1185	(51.9%)	8866	(38.3%)	**<0.001**
Postoperative	285	(12.5%)	7516	(32.5%)	**<0.001**
Multiple trauma	117	(5.1%)	819	(3.5%)	**<0.001**
Craniocerebral trauma	287	(12.6%)	896	(3.9%)	**<0.001**
Acute pancreatitis	94	(4.1%)	293	(1.3%)	**<0.001**
Obstetric complications	0	(0.0%)	87	(0.4%)	**0.006**
Acute neurological disorders	197	(8.6%)	1729	(7.5%)	0.053
Intoxication	148	(6.5%)	225	(1.0%)	**<0.001**
Severe metabolic disorders	299	(13.1%)	1095	(4.7%)	**<0.001**
Bacterial infection	414	(18.1%)	4412	(19.1%)	0.279
Sepsis	174	(7.6%)	1632	(7.1%)	0.342
Viral infection	6	(0.3%)	115	(0.5%)	0.163

**Table 3 ijerph-19-05914-t003:** ICU treatment in patients with or without AD.

Variables	Alcohol Dependent		Not Alcohol Dependent		*p*
	(n = 2285)		(n = 23,131)		
Use of catecholamines	1684	(73.7%)	16,894	(73.0%)	0.512
Intubation	1522	(66.6%)	14,942	(64.6%)	0.058
Tracheostomy	356	(15.6%)	3826	(16.5%)	0.249
Invasive ventilation	1991	(87.1%)	18,991	(82.1%)	**<0.001**
Non-invasive ventilation	62	(2.7%)	1132	(4.9%)	**<0.001**
Renal replacement therapy	249	(10.9%)	2472	(10.7%)	0.784
Operation while in the ICU	191	(8.4%)	2092	(9.0%)	0.292
Therapeutic hypothermia	31	(1.4%)	200	(0.9%)	**0.025**
Intra-aortic balloon pump	13	(0.6%)	638	(2.8%)	**<0.001**
Extracorporeal membrane oxygenation	4	(0.2%)	113	(0.5%)	0.051

**Table 4 ijerph-19-05914-t004:** Demographic data, admission source, and medical status at ICU admission among patients with AD—comparison of non-survivors and survivors.

		Non-Survivors		Survivors		*p*
		(n = 1092)		(n = 1193)		
Demographic data	Female sex	241	(22.1%)	202	(16.9%)	**0.002**
	Age >65 years	164	(15.0%)	155	(13.0%)	0.182
Admission	First	1056	(96.7%)	1133	(95.0%)	0.050
	Second	34	(3.1%)	58	(4.9%)	**0.044**
	Another	2	(0.2%)	2	(0.2%)	0.680
Admission source	Operating theatre	89	(8.2%)	137	(11.5%)	**0.009**
	Emergency department	436	(39.9%)	483	(40.5%)	0.818
	Surgical department	192	(17.6%)	197	(16.5%)	0.533
	Medical department	363	(33.2%)	364	(30.5%)	0.175
	Another ICU	12	(1.1%)	12	(1.0%)	0.990
Comorbidities	Coronary artery disease	171	(15.7%)	184	(15.4%)	0.922
	Heart failure	189	(17.3%)	155	(13.0%)	**0.005**
	Arterial hypertension	292	(26.7%)	334	(28.0%)	0.531
	Disseminated atherosclerosis	234	(21.4%)	171	(14.3%)	**<0.001**
	Chronic respiratory failure	71	(6.5%)	76	(6.4%)	0.966
	Home oxygen therapy	2	(0.2%)	1	(0.1%)	0.939
	Extreme obesity	19	(1.7%)	20	(1.7%)	0.964
	Cachexia	152	(13.9%)	101	(8.5%)	**<0.001**
	Diabetes	137	(12.5%)	133	(11.1%)	0.333
	Chronic renal failure	79	(7.2%)	41	(3.4%)	**<0.001**
	Dialysis dependency	7	(0.6%)	4	(0.3%)	0.452
	Previous cerebral stroke	39	(3.6%)	39	(3.3%)	0.778
	Chronic neurological disorders	107	(9.8%)	112	(9.4%)	0.793
	Systemic autoimmune diseases	2	(0.2%)	2	(0.2%)	0.680
	Post-transplant	1	(0.1%)	0	(0.0%)	0.965
	Cancer	14	(1.3%)	13	(1.1%)	0.817
	Pregnancy	0	(0.0%)	0	(0.0%)	-
	Other	205	(18.8%)	188	(15.8%)	0.064

**Table 5 ijerph-19-05914-t005:** Primary reasons for ICU admission among patients with AD—comparison of non-survivors and survivors.

Variables	Non-Survivors		Survivors		*p*
	(n = 1092)		(n = 1193)		
Acute respiratory failure	858	(78.6%)	864	(72.4%)	**0.001**
Exacerbation of respiratory failure	36	(3.3%)	46	(3.9%)	0.545
Exacerbation of circulatory failure	537	(49.2%)	320	(26.8%)	**<0.001**
Multiorgan failure	281	(25.7%)	92	(7.7%)	**<0.001**
Shock	460	(42.1%)	216	(18.1%)	**<0.001**
Cardiac arrest	480	(44.0%)	237	(19.9%)	**<0.001**
Disorders of consciousness	582	(53.3%)	603	(50.5%)	0.203
Postoperative	118	(10.8%)	167	(14.0%)	**0.025**
Multiple trauma	39	(3.6%)	78	(6.5%)	**0.002**
Craniocerebral trauma	115	(10.5%)	172	(14.4%)	**0.006**
Acute pancreatitis	53	(4.9%)	41	(3.4%)	0.090
Obstetric complications	0	(0.0%)	0	(0.0%)	-
Acute neurological disorders	85	(7.8%)	112	(9.4%)	0.197
Intoxication	49	(4.5%)	99	(8.3%)	**<0.001**
Severe metabolic disorders	201	(18.4%)	98	(8.2%)	**<0.001**
Bacterial infection	207	(19.0%)	207	(17.4%)	0.347
Sepsis	94	(8.6%)	80	(6.7%)	0.092
Viral infection	3	(0.3%)	3	(0.3%)	0.764

## Data Availability

The data presented in this study are available on request from the corresponding author. The data are not publicly available due to their sensitivity and legal restrictions.

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
