# Peer review of "Analysis of Patients with Alcohol Dependence Treated in Silesian Intensive Care Units"

_ijerph, 2022, doi:10.3390/ijerph19105914_

Round 1

Reviewer 1 Report

Smolarek et al. performed a retrospective  study in Poland  in 25,416 adult patients with alcohol dependence identified in a Silesian Registry of ICU. It is of interest to demonstrate that ICU treatment of patients with AD was associated with higher mortality and higher percentage of unfavourable outcomes, as some authors could have suggested it. Therefore, there is a question in which point the novelty of this paper is.

Comments;

  1. In the introduction, it is poor to use “At present, there are no data on ICU treatment of AD patients in Poland” as the rationale of this study. The authors need to clearly describe the clinical significance of this research topic and analyze the limitations of previous studies and knowledge gaps, to support the clinical needs of this study.
  2. Methods: From a methodological point of view a main outcome should be clearly stated as well as a main analysis and a necessary number of patients to be enrolled calculated.
  3. The criteria for AD should be detailed. 

Author Response

Comments and Suggestions for Authors

Smolarek et al. performed a retrospective  study in Poland  in 25,416 adult patients with alcohol dependence identified in a Silesian Registry of ICU. It is of interest to demonstrate that ICU treatment of patients with AD was associated with higher mortality and higher percentage of unfavourable outcomes, as some authors could have suggested it. Therefore, there is a question in which point the novelty of this paper is.

Comments;

  1. In the introduction, it is poor to use “At present, there are no data on ICU treatment of AD patients in Poland” as the rationale of this study. The authors need to clearly describe the clinical significance of this research topic and analyze the limitations of previous studies and knowledge gaps, to support the clinical needs of this study.

Answer:

We agree, that this rationale might be not sufficient, however we have already indicated the clinical significance of our research topic in the following statement: “Alcoholic patients have an increased morbidity and mortality in a postoperative period [6]. They also experience longer ICU and hospital length of stay, greater resource utilization, and significantly increased mortality compared to the remaining ICU population [7]. For all these reasons, population of patients with AD deserves special attention in the ICU setting.

Moreover, patients with AD are frequently admitted to the ICUs. The percentage of these patients varies across countries and is in the range from 7% to 33% [8]”.However, based on the Reviewer comment, we decided to  extend this description.

Action:

In the Introduction section, in lines 46-49, we have added the following sentences (they are marked in red):

„To the best of our knowledge, no studies based on a comparable number of data, analyzing the impact of alcohol dependence on survival in ICU patients have been published so far in the medical literature, and the majority of the already existing reports were single-centre studies”.

  • Methods: From a methodological point of view a main outcome should be clearly stated as well as a main analysis and a necessary number of patients to be enrolled calculated.

Answer:

Indeed, the primary aim seems to be too complex. We have limited it to the main comparison, as indicated below.

However, a necessary number of patients to be enrolled was not calculated. Instead, we decided to use the whole dataset available for these calculations.

Action:

A sentence in line 50-51: “Primary aim of our study was to describe population with AD treated in Polish ICUs, and to compare their pre-admission and admission variables as well as treatment and the outcomes with the remaining population, based on data from a local medical registry.”

was limited to:

„Primary aim of our study was to compare admission variables, the course of treatment and the outcomes of  ICU patients with AD with the remaining ICU population”.

  1. The criteria for AD should be detailed. 

Answer:

The criteria for AD are detailed in the Methods section: in the following sentence: “According to the Registry definitions (available via the Registry website) this variable may be introduced to the dataset if there is a clear information on the presence of AD in previous discharge cards or/and on the basis of the information obtained directly from the patient or his family.”

The Registry data were analyzed retrospectively. We are aware that during our data collection period (in 2013) the diagnostic criteria changed (in 2013), but this had no impact on the definitions used in the Registry. We are fully aware, that this explanation might not be satisfactory, but we were strictly limited to the data available in the Silesian ICU Registry.

At the time of data collection, the following definition of AD was valid:”Patient should be marked as AD, when there is a clear information on the presence of AD in previous discharge cards or/and on the basis of the information obtained directly from the patient or his family”. We fully agree, that this was a simplified definition, not based on classical diagnostic criteria, but only this definition was valid, accepted, and known to the Registry users,,when these data have been collected

Reviewer 2 Report

The manuscript titled "Analysis of patients with alcohol dependence treated in the Silesian intensive care units" by M. Smolarek is a report and analysis of ICU in a bit region of Polland. Anyway, the data introduced are interesting because few studies on alcohol impact in ICU.
I want to make some critical observations on the study limitations:
1. The method used to identify people with AD is not appropriate: it is possible that in the control group (no AD), there are many people with Alcohol Use Disorders (DSM 5), which confounding the results. I think to need a better description of the AD group (if possible). In another way, it's essential to write something about the possibility that alcohol-related pathologies can be present in no AD group.
2. Only approximately 35% of all Silesian ICUs reported their data to the registry at that time: the sample is not representative of Silesian people;
3. There are differences between gender in AD and not AD (it can influence the differences observed). I think it interesting to calculate the differences by gender;
4. In table 1: there are no illnesses typically correlated to alcohol like liver diseases; the number of cancers in the AD group is strange, considering alcohol is the second-factor causing cancer.  The difficulties of individualising AD in a retrospective study without appropriate screening tests can explain these data.

5. Statistical analysis: in some cases (admission from medical and surgical departments, …), it could be interesting to calculate not only the p-value but the effect size too.

Author Response

Answer:

It has been suggested (above in the table), that the conclusions of our study are not supported by the results. We have a different opinion. We concluded, that despite the younger age, ICU treatment of patients with AD was associated with higher mortality and higher percentage of unfavourable outcomes. We feel, that this statement is fully supported by our result. Population of patients with AD was significanly younger and the results of their treatment were significanly worse.

Comments and Suggestions for Authors

The manuscript titled "Analysis of patients with alcohol dependence treated in the Silesian intensive care units" by M. Smolarek is a report and analysis of ICU in a bit region of Poland. Anyway, the data introduced are interesting because few studies on alcohol impact in ICU.

I want to make some critical observations on the study limitations:

  1. The method used to identify people with AD is not appropriate: it is possible that in the control group (no AD), there are many people with Alcohol Use Disorders (DSM 5), which confounding the results. I think to need a better description of the AD group (if possible). In another way, it's essential to write something about the possibility that alcohol-related pathologies can be present in no AD group.

Answer:

As stated in our answer to a previous review, the Registry data were analyzed retrospectively, therefore they could be prone to bias. We were strictly limited to the data available in the Silesian ICU Registry. The criteria for AD are presented in the Methods section of our study in the following sentence: “According to the Registry definitions (available via the Registry website) this variable may be introduced to the dataset if there is a clear information on the presence of AD in previous discharge cards or/and on the basis of the information obtained directly from the patient or his family.” During our data collection period (in 2013) the diagnostic criteria for AD changed (in 2013), but this had no impact on the definitions used in the Registry.

At the time of data collection, the following definition of AD was valid:”Patient should be marked as AD, when there is a clear information on the presence of AD in previous discharge cards or/and on the basis of the information obtained directly from the patient or his family”. We fully agree, that this was a simplified definition, not based on classical diagnostic criteria, but only this definition was valid, accepted, and known to the Registry users,,when these data have been collected. Our Registry simply did not distinguish categories of “alcohol abuse” from “alcohol disorder”.

Therefore, it is possible indeed, that in our control group („no AD”), we might have included few patients with Alcohol Use Disorders (DSM 5) and classified as “Alcohol Abuse” (DMS-IV) or “Mild Alcohol Use Disorder (DSM-V). We have now stated it more clearly in the Discussion section in the limitations of the study.

Action:

In the Discussion section (lines 262-266), we have added the following two sentences:

“We used a simplified definition of AD, not based on classical diagnostic criteria, but only this definition was valid, accepted, and known to the Registry users, when these data have been collected. Therefore, it is possible indeed, that in our control group („no AD”), we might have included few patients with Alcohol Use Disorders (DSM 5) and classified as “Alcohol Abuse” (DMS-IV) or “Mild Alcohol Use Disorder (DSM-V).”

  1. Only approximately 35% of all Silesian ICUs reported their data to the registry at that time: the sample is not representative of Silesian people.

Answer:

Poland lacks a nationwide ICU registry. This deficit is partially filled, however, by the presence of a local registry operating in the Silesian Region of Poland since October 2010. To te best of our knowledge, this is also the only ICU Registry, operating in the Central-Eastern part of Europe.

Silesian ICU Registry however, is a voluntary registry. As such, this Registry is far from perfect and has several limitations. However, we are not aware of any significant differences between the ICUs that decided to participate and not to participate in the Registry. Also, we did not aspire to make our sample representative for the whole country or even a Silesian region. To our knowledge however, there is no study to date in the medical literature that has used similar methodology and comparable sample size.

  1. There are differences between gender in AD and not AD (it can influence the differences observed). I think it interesting to calculate the differences by gender.

Answer:

Percentage of male patients was significantly higher among patients with AD (80.6% vs 54%, p<0.001). Patients with AD were also significantly younger in comparison to the remaining population (mean age: 53.3 ±11.9 vs 62.2 ± 15.5 years, p<0.001).

The population of patients with AD was obviously different (in many aspects) from “no AD” patients. The differences in gender may have had an influence on the differences observed, but the same was also valid for the age differences and many other factors. We did not aim however, to calculate the differences by all these factors. Instead, we identified the differences in all these factors between the survivors and non-survivors (to achieve the primary aim of our study), and performed a multivariable analysis in search for independent risk factors for ICU death (to achieve the secondary aim of our study).

  1. In table 1: there are no illnesses typically correlated to alcohol like liver diseases; the number of cancers in the AD group is strange, considering alcohol is the second-factor causing cancer.  The difficulties of individualising AD in a retrospective study without appropriate screening tests can explain these data.

Answer:

We agree that it is true that AD is typically associated to cancer, but it has to be taken into account that our AD patients admitted to the ICU were a selected population – they were significantly younger, frequently admitted to the ICU following craniocerebral trauma, etc. Therefore, indeed, the difficulties of individualising AD in a retrospective study can explain these data. .

5.Statistical analysis: in some cases (admission from medical and surgical departments, …), it could be interesting to calculate not only the p-value but the effect size too.

Answer:

Effect size was calculated for the admission source.

Action:

Effect size has been added to comparisons in figure 2.

Reviewer 3 Report

Thank you for the opportunity to review the article of “Analysis of patients with alcohol dependence treated in the Silesian intensive care units”. This article is grossly written well with informative message. I had some comments for this article as:

  1. The aim of the study mentioned in the end of background is too rough and exploratory and can not reflect the merit of this study.
  2. Although the presence of AD or not is dependent on that registered in the system. The definition of the AD in this system should be mentioned and is it universally defined in the various institutes of the registry system?
  3. Define the minimally conscious state of the unfavourable outcome.
  4. In this study, there were multiple comparison of many variables simultaneously. Do the authors use the post hoc correction? and what kind of post hoc correction?
  5. As the author stated that the age contributing to the less chronic diseases of the selected population. Reasonably, the age is a main and stronger factor to the studied population and to the outcome. Since the control group is larger than the studied group, the author may consider the comparison in those age and gender – matched study groups to determine the impact on outcome.
  6. The discussion is too long and could be shortened. Of course, there would be some differences of two groups of patients, if you are doing more multiple comparison of the patients’ characteristics, associated illness, and outcome. You don’t need to explain them all, the author is encouraged on reporting the most important issues of the study (mortality or unfavourable outcome?), i.e., why you are doing this study to answer what question.
  7. The limitation section is too short and can not reflect the limitation of the study.
  8. Some spelling error should be corrected. For example, in the line 26 of Abstract, the space before “Despite” is too wide. In line 97 of Results, “25.416” should be “25,416”. In Table III, I would suggest “use of catecholamines” but not “Catecholamin”. Also, the full name of ECMO should be provided, it is familiar term by the CVS or trauma physicians but not all the readers from many fields. Table IV, “severe metabolic diserders" should be “disorders’. The spelling of the entire manuscript should be inspected again.

Author Response

Thank you for the opportunity to review the article of “Analysis of patients with alcohol dependence treated in the Silesian intensive care units”. This article is grossly written well with informative message. I had some comments for this article as:

1.The aim of the study mentioned in the end of background is too rough and exploratory and can not reflect the merit of this study.

Answer:

As mentioned in the answer to Reviewer 1, the primary aim seemed to be too complex. We have limited the aim to the main comparison, dividing it to the primary and secondary aim, as indicated below.

Action:

A sentence in line 50-51: “Primary aim of our study was to describe population with AD treated in Polish ICUs, and to compare their pre-admission and admission variables as well as treatment and the outcomes with the remaining population, based on data from a local medical registry.”

was limited to:

„Primary aim of our study was to compare admission variables, the course of treatment and the outcomes of  ICU patients with AD with the remaining ICU population”.

The secondary aim (“Secondary aim was to identify independent variables affecting ICU death in this specific population”) was left unchanged.

2.Although the presence of AD or not is dependent on that registered in the system. The definition of the AD in this system should be mentioned and is it universally defined in the various institutes of the registry system?

Answer:

Thank you very much for this comment. The similar issue has been already raised by another Reviewer.

As stated in the Methods section: „According to the Registry definitions (available via the Registry website) this variable may be introduced to the dataset if there is a clear information on the presence of AD in previous discharge cards or/and on the basis of the information obtained directly from the patient or his family”.Therefore, AD was defined in the identical way (provided in the Registry definitions) in the various institutes of the registry system.

Action:

We have added the following paragraph in the Discussion section (in the limitations of the study):

“We used a simplified definition of AD, not based on classical diagnostic criteria, but only this definition was valid, accepted, and known to the Registry users, when these data have been collected. Therefore, it is possible, that in our control group („no AD”), we might have included few patients with Alcohol Use Disorders (DSM 5) and classified as “Alcohol Abuse” (DMS-IV) or “Mild Alcohol Use Disorder (DSM-V)”.

3.Define the minimally conscious state of the unfavourable outcome.

Answer:

According to the Registry definitions (available via the Registry website, mandatory for all Registry users), patients have not been divided into patients in a vegetative state and in a minimally concious state (these two separate diagnoses could only be marked as one category). According to Registry definitions, patients in vegetative state or minimally concious state at ICU discharge was simply defined as “unconscious or with a minimal contact with the environment, functionally completely dependent on other people, requiring palliative care”.

4.In this study, there were multiple comparison of many variables simultaneously. Do the authors use the post hoc correction? and what kind of post hoc correction?

Answer:

We did not use post hoc corrections, as there was no indication for this technique. As stated in the Methods section, the effect of independent variables on the outcome variable of interest was first calculated by means of univariable logistic regression. Variables with p value <0.05 were included in multivariable logistic regression analysis. The multivariable model was then fitted using the stepwise method, where p<0.05 was set as inclusion and removal criteria.

5.As the author stated that the age contributing to the less chronic diseases of the selected population. Reasonably, the age is a main and stronger factor to the studied population and to the outcome. Since the control group is larger than the studied group, the author may consider the comparison in those age and gender – matched study groups to determine the impact on outcome.

Answer:

In our study, we did not aim to perform propensity score. As stated in the answer to previous reviewers, the population of patients with AD was obviously different (in many aspects) from “no AD” patients. The differences in gender may have had an influence on the differences observed, but the same was also valid for the age differences and many other factors. We did not aim however, to calculate the differences by all these factors. Instead, we identified the differences in all these factors between the survivors and non-survivors (to achieve the primary aim of our study), and performed a multivariable analysis in search for independent risk factors for ICU death (to achieve the secondary aim of our study).

  1. The discussion is too long and could be shortened. Of course, there would be some differences of two groups of patients, if you are doing more multiple comparison of the patients’ characteristics, associated illness, and outcome. You don’t need to explain them all, the author is encouraged on reporting the most important issues of the study (mortality or unfavourable outcome?), i.e., why you are doing this study to answer what question.

Answer:

We feel, that size of the Discussion section does not seem to be excessive. Having such a big dataset enabled us to present many clinically important aspects associated with the treatment process of AD patients in the ICU. Among them, there were circumstances surrounding ICU admission, assessment of comorbidities, treatment details and the outcomes. We aimed to put them in context, comparing our results with the data from the medical literature. It also seems that the most important issues of the study (mortality and unfavourable outcomes) are well discussed and confronted with the literature.  

7.The limitation section is too short and can not reflect the limitation of the study.

Answer:

We fully agree with this opinion. Therefore, we have added few sentences in the limitation section of our study.

Action:

As mentioned previously, we have added the following paragraph in the Discussion section (in the limitations of the study):

“We used a simplified definition of AD, not based on classical diagnostic criteria, but only this definition was valid, accepted, and known to the Registry users, when these data have been collected. Therefore, it is possible, that in our control group („no AD”), we might have included few patients with Alcohol Use Disorders (DSM 5) and classified as “Alcohol Abuse” (DMS-IV) or “Mild Alcohol Use Disorder (DSM-V)”.

8.Some spelling error should be corrected. For example, in the line 26 of Abstract, the space before “Despite” is too wide.

Answer:

Agree.

Action:

Corrected.

In line 97 of Results, “25.416” should be “25,416”.

Answer:

Agree.

Action:

Corrected.

In Table III, I would suggest “use of catecholamines” but not “Catecholamin”.

Answer:

Agree.

Action:

Corrected.

Also, the full name of ECMO should be provided, it is familiar term by the CVS or trauma physicians but not all the readers from many fields.

Answer:

Agree.

Action:

Corrected.

Table IV, “severe metabolic diserders" should be “disorders’. The spelling of the entire manuscript should be inspected again.

Answer:

Agree.

Action:

Corrected (in table V).

Round 2

Reviewer 1 Report

I think the paper has been properly answered and corrected for comments.

Reviewer 2 Report

No comment

Reviewer 3 Report

I have no further recommendation for the revised form of manuscript. Thank you.